# The Elaborated Assessment Framework of City Competitiveness from the Perspective of Regional Resource Integration

**Yin-Hao Chiu** [1,*] and **Yu-Yun Liu** [2]

1   Department of Urban Development, University of Taipei, Taipei 11153, Taiwan
2   Bachelor of Leisure Management, Chinese Culture University, Taipei 11114, Taiwan; yuyliu@sce.pccu.edu.tw
*   Correspondence: yhchiu@go.utaipei.edu.tw; Tel.: +886-02-2871-8288 (ext. 3102)

**Abstract:** Studies on regional development have generally focused on major cities, to the neglect of minor ones. In this research, a secondary city in Taiwan, namely Keelung, was selected as a case study for urban development assessment from the perspective of regional resource integration. This study combined the decision-making trial and evaluation laboratory (DEMATEL) method and analytic network process (ANP) to determine how dimensions influenced each other in Keelung and what their weights were. We used six dimensions that comprised 14 criteria. The adopted dimensions were economics, governance, society, physical environment, natural environment, and culture and creativity. On the basis of the DEMATEL-based ANP, experts considered the governance, economics, and society dimensions to be highly important and influential on the other dimensions. In elucidating the relationships between dimensions, our method allows policymakers to formulate holistic solutions rather than piecemeal ones. The satisfaction of experts and residents (who have expertise and considerable living experience in Keelung, respectively) with the current situation in Keelung regarding the dimensions and criteria was also determined.

**Keywords:** regional integration; city competitiveness; multiple criteria decision-making (MCDM); decision-making trial and evaluation laboratory-based analytic network process (DANP); urban study; urban policy

## 1. Introduction

The concept of an urban area, which was formulated in the 1950s, was originally considered an important analytical tool in the United States and was then increasingly applied in Europe from the 1980s onward; it has also made urban development more complex [1]. In the context of the formation and integration of development policies in the 1980s, old concepts of space became unable to reflect the global reach of capitalist development or the rapid changes occurring in various regions. Thus, these old concepts were insufficiently global yet also insufficiently local (to reflect local demands). Brenner (1998) [2] believed that under a capitalist regime in the 1980s, changes in the form of national organization in global cities represented a transition toward complete globalization in which the superimposed spatial scale was reconfigured. The geographical environment during the aforementioned period could be considered as a collection of multilevel and multiform local organizations that included cities, intercity networks, and territorial states—all of which overlapped with each other and were becoming increasingly intertwined.

Regional policies have been formulated in response to the problems of sociospatial polarization and uneven geographical development caused by neoliberalism [3]. Prior to the 1980s, regional integration revolved around a core city in the region and interactions occurred in relation to the surrounding hinterland, in response to industrial imperatives, and in a manner that reflected geographical relationships. Furthermore, because of an intensification of resource exchange and technical mobility, administrative boundaries have been blurred. Consequently, greater demands for regional development, resource

distribution, and reorganization in governance have been made. However, noncore cities have been neglected, which has resulted in regional imbalance that has caused such cities to become sidelined, become prone to recession, and be allocated a disproportionately low amount of resources.

In this study, Keelung City (Keelung) was analyzed to validate the aforementioned claims. Keelung was once the most important (and thus prosperous) harbor city in northern Taiwan. However, Keelung declined in importance as the economy evolved and surrounding cities overtook it. This situation has been compounded by a lack of clarity in Keelung's development direction. Therefore, this study analyzed Keelung's situation to formulate several recommendations related to its development direction.

This study used indices of urban development and competitiveness to analyze the situation in Keelung. The different roles played by different cities were considered and an "index system" was incorporated to support the decision-making process pertaining to urban development strategies. The goal was to explore and analyze theories of city positioning and function in the context of Keelung, which is characterized by increasingly tight regional cooperation. Subsequently, the indices were used to identify the importance of different cities in regional development, and an assessment framework was established to determine the status of urban development and regional resource allocation. Thereafter, recommendations related to the development direction of Keelung were formulated. The evaluation of city competitiveness was holistic, focusing not only on the economy but also on governance, society, the built environment, the natural environment, culture, and innovation, which is congruent with the concept of sustainable development. The formulated recommendations can elevate the overall competitiveness of a region and help it develop sustainably.

## 2. Theory

### 2.1. Cooperation and Competition between City Regions

Hass proposed neofunctionalism, which is based on traditional regionalism. Hass's idea centers on the spillover effect that occurs during the regional integration process. Such an effect involves parts of a region working together for their common good; a region may even be involved with issues outside its traditional scope of concern or work with other regions situated at different levels [4]. Such spillover effects are also present in cities. In general, regional complementarity and cooperation make a city more competitive. Specifically, the increased wealth of core cities in a region spills over to surrounding suburbs and thus benefits the region as a whole [5].

The theory of neoregionalism emerged in the late 1980s when regionalism failed to explain the patterns of cooperation present in countries and regions outside of Europe [6]. Sun (2003) [7] defined neoregionalism as multilevel integration in the form of regional free trade networks or national security alliances that emphasizes regional cohesiveness and a common political philosophy. Globalization has given rise to the phenomenon of spatial flow and to changes in regions' economic structure, social environment, and power structure, which has changed the traditional sense of spatial scale [2]. More autonomous local governments, a more robust civil society, and greater human migration due to urbanization have changed how national land space is used. Because of globalization after 1980, many countries' integrated development policies have often been unable to cope effectively with capitalism-induced problems in their development and with the expansion of the world system. These policies have also been unable to respond to the rapidly changing local demands in parts of a region. Thus, the scale of the aforementioned policies is not only insufficiently global but also insufficiently local. Neoregionalism has thus emerged in response to this problem. The theory of neoliberalism advocates the formation of functional metropolitan regions and the pursuit of common interests through regional cooperation in the global network. In studies on region-oriented spatial planning, Storper (1996) [8], Morgan (1997) [9], and Keating (1998) [10] have found that the "city region" is the best scale for realizing network development because such a region is small enough to enable

face-to-face interactions (which build trust and facilitate cooperation) and large enough to support key interpersonal and industrial networks. Castells (2010) [11] argued that the key features of urbanization are the diffusion and networking of metropolitan areas. Chou and Chen (2014) [12] stated that the formation of metropolitan areas entails a network-based development involving the flow of people, material, capital, information, and technology and thus the formation of a new space. However, such development creates challenges for multidimensional and multiscale governance.

Asia features many examples of the city region developmental model, including the Greater Tokyo Area (Shutoken) in Japan. The Shutoken includes one metropolis and seven prefectures (the Tokyo Metropolis and the Kanagawa, Chiba, Saitama, Gunma, Tochigi, Ibaraki, and Yamanashi Prefectures), which cover approximately 45,007 km$^2$ or 9.8% of Japan's land area. The aforementioned region has a population of approximately 45.3 million, which accounts for 35.64% of Japan's population [13]. Japan began planning the development of Shutoken in the 1950s and has just implemented its fifth plan for the region. With regard to scope, except for the first plan, the other four plans for Shutoken covered an area named "one metropolis, seven prefectures." The first Shutoken plan began with the objective of controlling the expansion and spread of the Tokyo Metropolis. Subsequently, the plan's objective gradually shifted toward reducing regional gaps and promoting development in surrounding areas. In 1999, the objective of the fifth Shutoken plan was to enhance regional competitiveness; promote sustainable development; and implement a core city as a center that has a self-reliant, complementary, and high-intensity "decentralized network area spatial structure" that is relatively independent and can facilitate exchange [14].

Good results followed from Shutoken's transformation from a "monopole core" network model to a "multipole multicore" network model. For example, the population spread from core cities to surrounding regions, and the trend of a multicenter network with functional organization took hold in Shutoken. Through the exercise of their "geoeconomic" advantages, the surrounding areas became critical functional areas and growth points of Shutoken.

The working objective of the half-century-old networked spatial organization of Shutoken was to plan functional sites, such as multicircle and diversified subcity centers and satellite cities, according to the comparative resource advantages and locations of different regions for developing reasonable guidelines for the orderly discharge of functions. The first tier of suburban cities was distributed within 10 km of the main urban area. A clear functional division enabled these cities to assume jointly the core functions of the capital. The second tier of commercial centers was located within the 1-h-commute circle from the main urban area and within 10–70 km from central Tokyo. The urban functions of the second tier of cities included administration, commerce, area logistics, and secondary administration. The aforementioned two tiers constituted the core functional areas of the Tokyo Metropolis. The third tier of nuclear cities were located within 70–150 km from central Tokyo and comprised centers for industry, logistics, and research, thereby providing robust support for the capital's development [15,16]. The objectives and course of development of Shutoken indicates that Japan attaches considerable importance to the various functional roles of urban areas. Through the division of labor and on the basis of the complementarity-based developmental model, urban areas around Tokyo jointly support its developmental demands and enhance the competitiveness of the overall region.

## 2.2. Urban Development and Competitiveness

The concept of competitiveness was initially used to refer to the competition between businesses [17,18] and was primarily determined by a business's profitability. However, when neoliberalism emerged in the 1980s, the notion of competitiveness began to be applied to cities, countries, and regions. Neoliberals believe that a city's competitiveness is determined by its economic performance [19–21]. Kresl (1995) [20] conceptualized competitiveness as being driven by only economic and strategic factors. This view has

resulted in the neglect of other factors, such as environmental friendliness, in the discussion on competitiveness. However, cities possess values that span multiple dimensions. A competitiveness report released by the World Economic Forum in 2015 indicated that to handle the new norms formed after the 2008 global financial crisis, many countries and cities began to accept the concept of inclusive growth, which emphasizes simultaneous and equal growth in all dimensions of urban development (WEF, 2015) [22]. Wang (2009) [23] used Red Town (a creative park in Shanghai) as an example to illustrate how a decision-making unit's singular focus on economic development led to unforeseen social problems. Thus, assessments of a city's competitiveness cannot solely be based on its short-term economic success, and social and environment dimensions must also be considered.

### 2.3. Development of Keelung City and Taipei Metropolitan Area

Keelung City is located in the north of Taiwan, and it comes under the Taipei metropolitan area together with Taipei City and New Taipei City (Figure 1). Keelung has an area of 132.7589 km$^2$, and 58.3% of this area is hilly terrain that is no taller than 50 m above sea level. Only 25% of the land in Keelung (primarily valleys and coastal areas) is suitable for (primarily port-related) development [24]. The rapid industrialization of Taiwan in the 1970s led to the emergence of a metropolitan area in Taipei and cemented Taipei's status as the primary metropolis in Taiwan. Under the 2010 Strategic Plan for National Spatial Development, which aimed to enhance the nation's competitiveness, Taipei was designated as the primary gateway to the rest of the country, the core of Taiwan's trade and economy, a global city of research and culture, and a space for high-tech industry.

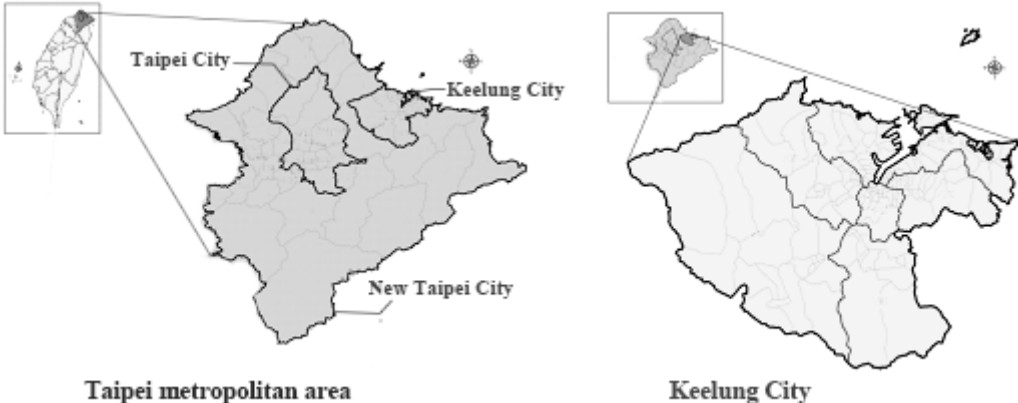

**Figure 1.** Spatial relationship diagram of Keelung City and Taipei metropolitan area.

However, the process of globalization in Keelung has not been homogeneous. Keelung used to be the primary harbor city in the Taipei metropolitan area; however, in recent years, it has been beset by a host of problems, such as population outflows, economic recessions, and reduced employment, as indicated in increases in indices of social problems (including the unemployment rate) and a decrease in household disposable income. Furthermore, national policy has focused on the development of the Taipei metropolitan area, to the neglect of Keelung, because of the expanding size and economic clout of this metropolitan area (Executive Yuan, 2016) [24], which have caused a regional imbalance in development. Moreover, if competitiveness is assessed on a city-by-city basis, Keelung City may be cast as being uncompetitive and unworthy of investment.

Therefore, this study addressed the drawback of conceptualizing competitiveness at only the city level by adopting the perspective of regional resource integration to reconfigure the allocation of resources within a region. Making Keelung more competitive or functionally useful benefits not only Keelung but also enhances resource integration and benefits the region as a whole.

## 3. Methods

This study used multiple criteria decision-making, the fuzzy Delphi method (FDM), and the decision-making trial and evaluation laboratory (DEMATEL)-based analytic network process (ANP) to explore the urban development strategy of a secondary city. Expert questionnaires were used to obtain data. The results of the DEMATEL-based ANP (DANP) indicate the connections between various criteria and thus inform decision-making.

### 3.1. The Fuzzy Delphi Method

The FDM evolved from combining the conventional Delphi method and fuzzy set theory. The Delphi method is tedious and time-consuming to implement, and the questions in the questionnaire might be semantically confusing, thus causing erroneous responses; as such, subsequent scholars improved the drawbacks of the conventional Delphi method. Ishikawa et al. (1993) [25] incorporated fuzzy set theory into the Delphi method to overcome these problems and employed the concepts of cumulative frequency distribution and fuzzy integral to transform the expert opinions into fuzzy numbers, through which the FDM was developed. Jeng (2001) [26] employed the double triangular fuzzy technique and gray zone test method to examine and integrate expert opinions and to reduce the frequency of repeated questionnaire surveys. Since these approaches are more objective and reasonable than the single triangular fuzzy technique for obtaining the geometric mean, the present study developed the relevant criteria and indices for assessing the secondary city. Normally, the FDM comprises the following four steps of inspection:

Step 1: Ask all the experts to assign possible interval values for each assessment item.

Step 2: For each assessment item (i.e., assessment item $i$), record the "most conservatively perceived values" and the "most optimistically perceived values" given by all the experts and remove extreme values that fall outside of two standard deviations. Next, out of the remaining values, calculate the minimum value ($C_L^i$), geometric mean ($C_M^i$), and maximum value ($C_U^i$) of the "most conservatively perceived values" as well as the minimum value ($O_L^i$), geometric mean ($O_M^i$), and maximum value ($O_U^i$) of the "most optimistically perceived values."

Step 3: For each assessment item (i.e., assessment item $i$) calculated in Step 2, record the triangular fuzzy number for the "most conservatively perceived values" (i.e., $C^i = \{C_L^i, C_M^i, C_U^i\}$ and that for the "most optimistically perceived values" (i.e., $O^i = \{O_L^i, O_M^i, O_U^i\}$).

Step 4: Determine whether the experts reach a consensus in their opinions:

(1) If the two triangular fuzzy numbers do not overlap (i.e., $C_U^i < O_L^i$), it means that a consensus interval (or consensus intervals) exists in the experts' opinion intervals and that their opinions will likely fall within the consensus interval range. Thus, the "value concerning the importance of consensus" ($G^i$, hereafter referred to as "consensus importance value") of assessment item $i$ is equal to the arithmetic mean of $C_M^i$ and $O_M^i$; in other words, $G^i = (C_M^i + O_M^i)/2$.

(2) If the two triangular fuzzy numbers overlap (i.e., $C_U^i > O_L^i$) and the fuzzy relationship gray area ($Z^i = C_U^i - O_L^i$) is smaller than the interval range assigned by the experts for the assessment item (i.e., $M^i = O_M^i - C_M^i$, which represents the interval range between the "geometric mean of the optimistically perceived values" and the "geometric mean of the conservatively perceived values), then although no consensus intervals exist in the experts' opinion intervals, the extreme values (i.e., the most conservative value in the optimistically perceived values and the most optimistic value in the conservatively perceived values) given by two experts do not differ significantly from the opinions of other scholars; accordingly, no divergence of opinion has occurred. Therefore, the consensus importance value of assessment item $i$ (i.e., $G^i$) is ordered to be equal to the fuzzy set obtained by computing the fuzzy

relationships of the two triangular fuzzy numbers. Next, the quantitative score, which contains the maximum membership degree of the fuzzy set, is calculated.

$$F^i(X_i) = \left\{ \int_x \left\{ \min \left[ C^i(X_i) \right], O^i(X_i) \right. \right. \tag{1}$$

$$G^i = \left\{ \left\{ X_i \middle| \max \mu_{p^i} \right. \right. \tag{2}$$

(3)     If the two triangular fuzzy numbers overlap (i.e., $C_U^i > O_L^i$) and the gray area of fuzzy relationship ($Z^i = C_U^i - O_L^i$) is greater than the interval range assigned by the experts for the assessment item (i.e., $M^i = O_M^i - C_M^i$, which represents the interval range between the "geometric mean of the optimistically perceived values" and the "geometric mean of the conservatively perceived values), then no consensus intervals exist in the experts' opinion intervals, and that the extreme values (i.e., the most conservative value in optimistically perceived values and the most optimistic value in conservatively perceived values) given by two experts differ significantly from the opinions of other scholars; accordingly, a divergence of opinion has occurred. Thus, assessment items containing opinions that do not converge are provided to experts for review and Steps 1 to 4 are repeated for another round of questionnaire survey until all assessment items converge and the consensus importance value (i.e., $G^i$) is obtained.

### 3.2. The Decision-Making Trial and Evaluation Laboratory-Based Analytic Network Process

The decision-making trial and evaluation laboratory-based analytic network process (DANP) is a composite method used in multi-criteria decision analyses. The influential weight of such method is derived by applying the analytic network process method (ANP) to the decision-making trial and evaluation laboratory method (DEMATEL). The DEMATEL method was to build an interconnected network pattern and structure to analyze complex real-world situations [27]. By adopting a hierarchical structure, the said method effectively resolves complex, convoluted social problems and identifies the causal relationships between different events; the method is thus used to research the handling of complex events [28]. The DEMATEL method uses the specific characteristics of target events to prove the correlations between the characteristics as well as how much they affect one another [29,30].

When using the ANP method, a correlation matrix obtained using the DEMATEL is first used to calculate weights. Next, a search is made to identify the key criteria. Lee et al. (2009) [31] applied the total influence matrix of DEMATEL to ANP supermatrices to determine the effect of dynamic importance (as opposed to traditional importance). Because the factors in the present study were highly correlated, the DANP method was chosen to obtain their global influential weights.

### 3.3. Obtain the DANP Influential Weights

The ANP method was derived from the analytic hierarchy process (AHP) method proposed by Saaty in 1980 [32]. Similar to the AHP method, the ANP method is used when making multi-criteria decisions; however, the ANP method features more relaxed restrictions on hierarchical structure [33]. Concerning the major difference between the two methods, it is described as follows: the ANP method is used in decisions in which the projects or criteria mutually influence one another, whereas the AHP method is used in decisions in which the projects or criteria are mutually independent. Therefore, to understand the influences between factors, the ANP method is considered more appropriate. In this study, the DEMATEL method was used to determine the relationships between the dimensions and the indices, and influential weights were obtained to understand the problems to be solved.

The ANP method mainly considers the limitations of the AHP method (the hierarchy elements are presumed to be independent) and through the interdependency and regression concept, considers interdependency in the weight calculation. Through comparison of the super matrix, which is composed of weight priority matrices, the results are obtained. Using the ANP to assess the two cases to understand the difference between them could further clarify that when interdependency is present among cases or criteria, the ANP is more suitable than the AHP method as a case selection tool [34,35]. Interdependency characteristics can be classified into the following three types:

(1)  Technical interdependency: The factors behind success or failure in the mutual development of mutual influence between two schemes.

(2)  Resource interdependency: When a resource developed by one scheme is applied to another scheme, the two schemes share interdependent or interest relationships for an organization in terms of benefits.

(3)  Benefit interdependency: For an organization, the implementation of two interdependent schemes could enhance the expected result; therefore, considering the interdependency characteristic between the schemes could enable cost reduction or profit generation for an organization.

Therefore, the interdependency of the ANP could be closely related to the human thought process, and a decision maker could determine the optimal decision according to the actual situation. This method has greater value in decision-making in most real situations; its application includes weight sorting, alternate product and plan assessment, forecasting strategic decisions, and risk assessment [36]. In addition, its application area is very wide, including education, construction, finance, marketing, information management, industrial management, decision sciences, and health care administration. In recent years, many studies have applied the ANP for research related to urban and regional issues. Grimaldi et al. (2017) [37] used the ANP to analyze water infrastructure plans. Malmir et al. (2016) [38] analyzed land suitability for the urban development of Ahwaz in Iran, covering biophysics and socioeconomic dimension criteria in their assessment and marking the area suitable for urban development. Zhang and Wang (2015) [39] used the ANP to analyze and evaluate city and river ecosystem proposals in Henan Province, China, and through data analysis designed an appropriate proposal that the researchers believed could also be applied to improve urban green space planning. Javadian Koutenaee et al. (2014) [40] used the ANP to design an assessment model to assess ecological capacity during urban development, used the analysis results to propose land-use zoning based on the ecological capacity of urban development, and attempted to reduce the environmental impact. Jeong et al. (2014) [41] proposed increasing urgency in urban prevention planning and emphasized that focus must be on the assessment indicators of preventive roles related to urban safety and natural disasters Therefore, the researchers used the ANP to construct compact urban indicators that reflected preventive planning factors.

## 4. Results

*Urban Development Assessment Indices*

The present study constructed an urban development assessment system by modifying similar systems in the literature according to the aforementioned dimensions, a literature review, and the opinions of experts (obtained from interviews). Dimensions that recurred were regarded as being more important and thus retained, and new dimensions were constructed where necessary.

The following six major dimensions (and 26 criteria) were obtained from a literature review (Table 1): economics (five criteria), governance (four criteria), society (four criteria), the physical environment (four criteria), the natural environment (four criteria), and culture and creativity (four criteria). The FDM was used to choose which dimensions to retain according to relevance as per expert opinion.

**Table 1.** Statistical analysis and selection results obtained by using the urban development competitiveness.

| | Dimensions | Statements of Influence Criteria |
|---|---|---|
| **1. Economics** | C1. Total output | Total output from all industries in a city |
| | C2. Growth rate of total output | Annual growth rate of total output from all industries in a city |
| | C3. Employment rate of the labor force | Employment rate of the working-age population, which reflects the number of job opportunities in a city |
| | C4. Per capita income | Per capita income of the urban residents |
| | C5. Import and export trade volumes | The volume of goods imported in and exported from of a city's ports and airports, which reflects the city's importance in international trade |
| **2. Governance** | C6. Government efficiency | The efficiency of the local government units when implementing policies (e.g., whether governance failure occurred because of excessive delays) |
| | C7. Monitoring mechanism | The completeness of local governments' monitoring mechanisms, which involve the between-agency monitoring among government agencies and public participation to prevent government agencies from acting solely on their own volitions |
| | C8. Public participation | Whether sufficient participation mechanisms are in place for the public to respond to its local governments' governance objectives and attitude (measured by means such as voting rates and how satisfied the public is with the channels in which they can voice their opinions) |
| | C9. Cross-border cooperation | All cross-border cooperation matters such as collaborative policies and constructions. Methods for assessing the local governments' border governance abilities include the number of projects taken and the local governments' implementation performance |
| **3. Society** | C10. Education level | Urban residents' education level (assessed mostly by bachelor's and master's degree) constitutes an influencing factor of the human resource quality of a city |
| | C11. Standard of living | The city's quality of life, residents' satisfaction level, and the extent to which the residents identify themselves with their communities are adopted to show the subjective and objective evaluations of the city's standard of living |
| | C12. Social security | Measured mostly by a city's public order, crime rate, and death rate by crimes |
| | C13. Social welfare | Examples include care policies for children, older adults, and/or other disadvantaged groups as well as insurance and/or other subsidies for residents |
| **4. Physical environment** | C14. Quality of infrastructure | The extent to which the infrastructure (e.g., sewers, roads, and networks) is completed and how satisfied the public is with such infrastructure |
| | C15. Public facilities | The average area of public facilities that each person can use (the public facilities are defined by a region's urban planning law) |
| | C16. Medical resources | The amount of medical resources available for the residents, including the average number of hospital beds and medical aids available per person |

**Table 1.** *Cont.*

| Dimensions | | Statements of Influence Criteria |
|---|---|---|
| | C17. Public transport | The capacity of, utilization rate of, and residents' satisfaction with the city's public transport (e.g., buses, mass rapid transits, and public bicycles) and whether the city has enough public transport facilities to support its residents |
| 5. Natural environment | C18. Pollution index | Assessment of the city's environment quality, which is measured by the city's water, air, and waste pollution situations |
| | C19. Energy consumption | Assessment of the city's energy consumption according to energy consumption rates of its households and factories |
| | C20. Green space ratio | Ratio of green coverage in the city (the size of the city is defined by the size of the city as stated in its urban project) |
| | C21. Renewable energy usage rate | Renewable energy consumed as a ratio of the total energy consumed |
| 6. Culture and creativity | C22. Cultural and creative industries | The number of people who are involved with the cultural and creative industries as well as the number of job opportunities offered by the said industry. The number of related personnel working in the cultural and creative industries (which consist of the 16 industries selected by the Ministry of Culture) as a ratio of the total employment population of the city |
| | C23. Material cultural heritage | The quantity of material cultural heritage, which can serve as the city's unique features to attract outside visitors as well as an important source of inspiration for developing the city's cultural and creative industry. Material cultural heritage primarily consists of monuments, historical buildings, settlements, and ruins that have been registered in the database of Ministry of Culture |
| | C24. Nonmaterial cultural heritage | The quality of the nonmaterial cultural heritage, which includes unique cultural assets in people's lives and mainly comprises cultural landscapes, traditional arts, and folk activities registered in the database of Ministry of Culture |
| | C25. Number of arts and cultural activities held | The development of the cultural and creative industries relies on favorable development environments; art and cultural activities can popularize cultural education, enhance people's overall cultural literacy, and foster people's creation and appreciation abilities |
| | C26.Tourism attractiveness | The number of tourists in the city's main tourist attractions, which reflects the popularity of the city's tourism industry |

On the basis of the literature review in the Theory section, the present study selected criteria for measuring urban development and competitiveness and used the FDM to select which criteria to use. Dalkey and Helmer (1963) [42] suggested that experiments with at least 10 evaluations have an acceptable reliability and error rate. Although Delbecq et al. (1975) [43] argued that a sample of 15–30 participants is acceptable for a Delphi study if the sample is homogeneous, a sample of 5–10 experts can be used for a study that involves various evaluations. Considering these methodological recommendations and the context of this study, we invited nine experts to complete the FDM expert questionnaire.

The participants were experts in architecture, urban planning, transportation, or disaster prevention with at least 20 years of experience. Finally, 14 out of 26 criteria that had an expert consensus value of >6.5 in the questionnaire were retained (Figure 2 and Table 2). The economics, governance, society, physical environment, natural environment, and culture and creativity dimensions had two, two criteria, three criteria, four criteria, one criterion, and two criteria, respectively. Subsequently, we used the DANP to determine the relationships between the criteria.

**Table 2.** Statistical analysis and selection results obtained by using the local industry development indices.

| Assessment Indices | Minimum Value Ci | | Maximum Value, Oi | | Single Value, a | | Geometric Mean | | | Expert Consensus Value Gi | Whether to Keep the Index (Gi > 6.5) |
|---|---|---|---|---|---|---|---|---|---|---|---|
| | Min | Max | Min | Max | Min | Max | Ci | Oi | Single Value | | |
| C1. Total output | 3 | 7 | 7 | 10 | 5 | 8 | 4.871 | 8.254 | 6.451 | 6.562 | Keep |
| C2. Growth rate of total output | 3 | 7 | 6 | 10 | 4 | 8 | 4.707 | 8.235 | 6.090 | 6.494 | |
| C3. Employment rate of the labor force | 2 | 7 | 5 | 10 | 5 | 8 | 5.188 | 8.310 | 6.946 | 6.292 | |
| C4. Per capita income | 4 | 7 | 6 | 8 | 7 | 10 | 5.798 | 8.615 | 7.285 | 7.216 | Keep |
| C5. Import and export trade volumes | 3 | 6 | 6 | 9 | 4 | 7 | 4.372 | 7.149 | 5.780 | 5.760 | |
| C6. Government efficiency | 3 | 8 | 7 | 10 | 4 | 9 | 5.474 | 8.713 | 6.952 | 7.404 | Keep |
| C7. Monitoring mechanism | 3 | 7 | 6 | 9 | 5 | 8 | 4.610 | 7.590 | 6.263 | 6.400 | |
| C8. Public participation | 3 | 7 | 5 | 9 | 4 | 8 | 4.705 | 7.680 | 6.327 | 6.077 | |
| C9. Cross-border cooperation | 4 | 7 | 7 | 10 | 6 | 8 | 5.603 | 8.503 | 7.072 | 7.053 | Keep |
| C10. Education level | 4 | 6 | 7 | 10 | 6 | 7 | 5.183 | 8.066 | 6.425 | 6.625 | Keep |
| C11. Standard of living | 5 | 7 | 7 | 10 | 6 | 8 | 5.726 | 8.728 | 7.194 | 7.227 | Keep |
| C12. Social security | 2 | 8 | 6 | 10 | 5 | 9 | 4.838 | 7.915 | 6.628 | 6.754 | Keep |
| C13. Social welfare | 3 | 7 | 7 | 9 | 5 | 8 | 4.671 | 7.972 | 6.356 | 6.322 | |
| C14. Quality of infrastructure | 5 | 7 | 8 | 10 | 7 | 8 | 5.726 | 8.975 | 7.319 | 7.350 | Keep |
| C15. Public facilities | 3 | 8 | 6 | 10 | 5 | 9 | 4.741 | 7.799 | 6.440 | 6.711 | Keep |
| C16. Medical resources | 4 | 8 | 7 | 10 | 5 | 9 | 5.287 | 7.933 | 6.572 | 7.256 | Keep |
| C17. Public transport | 5 | 8 | 8 | 10 | 7 | 9 | 6.033 | 9.188 | 7.526 | 7.610 | Keep |
| C18. Pollution index | 4 | 7 | 6 | 10 | 5 | 8 | 5.018 | 8.370 | 6.712 | 6.545 | Keep |
| C19. Energy consumption | 3 | 7 | 6 | 9 | 4 | 8 | 4.762 | 7.603 | 5.803 | 6.417 | |
| C20. Green space ratio | 3 | 7 | 6 | 9 | 5 | 8 | 4.882 | 7.398 | 6.052 | 6.398 | |
| C21. Renewable energy usage rate | 3 | 7 | 6 | 9 | 5 | 8 | 4.692 | 7.509 | 6.248 | 6.395 | |
| C22. Cultural and creative industries | 4 | 9 | 7 | 10 | 3 | 8 | 5.726 | 8499 | 6.024 | 7.628 | Keep |
| C23. Material cultural heritage | 3 | 7 | 6 | 9 | 5 | 8 | 4.389 | 7.478 | 6.000 | 6.362 | |
| C24. Nonmaterial cultural heritage | 2 | 7 | 6 | 9 | 5 | 8 | 4.500 | 7.690 | 6.336 | 6.403 | |
| C25. Number of arts and cultural activities held | 2 | 7 | 6 | 9 | 4 | 8 | 4.196 | 7.256 | 5.669 | 6.309 | |
| C26. Tourism attractiveness | 4 | 6 | 6 | 10 | 6 | 8 | 5.621 | 8.843 | 7.194 | 7.232 | Keep |
| Total number of criteria selected: 14 | | | | | | | | | | 6.500 | |

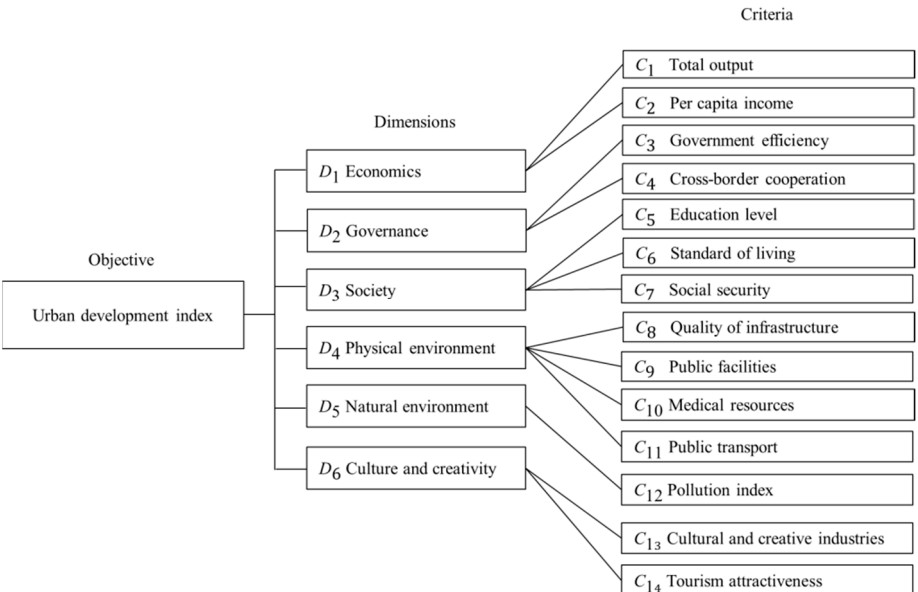

**Figure 2.** Urban development competitiveness dimensions and criteria.

The criteria that the experts considered to be more relevant were obtained using the FDM, after which the effects that the criteria exerted on each another were identified using the DEMATEL method. Previous studies have mostly investigated the performance of individual assessment indices and have failed to illustrate systemic problems. Therefore, the present study employed the DANP to determine the weights of the criteria and their effects on each other for identifying pre-existing problems to formulate relevant policy recommendations.

The mean of the direct relation matrix (i.e., D) was used to create the initial influence matrix (i.e., A). We interviewed seven experts in urban planning, transportation, disaster prevention, or architecture with at least 10 years of experience. The initial influence matrix A was obtained from the questionnaire data. The degree of influence was obtained by subtracting $s_i$ from $r_i$. A given criteria influenced other criteria if $s_i - r_i > 0$; conversely, a criteria was influenced by other criteria if $s_i - r_i < 0$. The aforementioned conditions were used to determine the interrelationships between the criteria. The total degree of influence of the dimensions and criteria is presented in Table 3.

According to the results, the experts believed that the economics dimension ($D_1$) exerted and received the most influence (centrality: 2.246), followed by the dimensions of culture and creativity $D_6$; centrality: 2.191), society ($D_{13}$), the physical environment ($D_4$), governance ($D_2$), and the natural environment ($D_5$). In addition, the dimensions with the highest to lowest $s_i - r_i$ values were governance ($D_2$), society ($D_3$), economics ($D_1$), the natural environment ($D_5$), the physical environment ($D_4$), and culture and creativity ($D_6$). Thus, among all dimensions, governance exerted more influence than it received and should thus be a policy priority.

The map of influence within the network reveals the experts' assessments of the influence of the dimensions on urban competitiveness. The governance dimension was assessed to be the most influential (Table 3), and governments should, for example, improve administrative efficiency or cross-border cooperation. The society dimension was the second most influential on a city's overall development, and governments should, for example, improve the citizens' quality of life or strengthen welfare policies. Measures such as the aforementioned ones are crucial because a good living environment is foundational to the success of a city.

**Table 3.** The sum of influences given and received on dimensions and criteria.

| Dimensions and Criteria | | $r_i$ | $s_j$ | Centrality (Prominence) $r_i + s_j$ | Degree of Influence (Relation) $r_i - s_j$ |
|---|---|---|---|---|---|
| $D_1$ | **Economics** | 1.161 | 1.085 | 2.246 | 0.076 |
| $C_1$ | Total output | 2.524 | 2.768 | 5.292 | −0.244 |
| $C_2$ | Per capita income | 3.020 | 2.325 | 5.345 | 0.696 |
| $D_2$ | **Governance** | 1.199 | 0.864 | 2.063 | 0.335 |
| $C_3$ | Government efficiency | 2.927 | 1.892 | 4.819 | 1.035 |
| $C_4$ | Cross-border cooperation | 2.835 | 2.213 | 5.048 | 0.622 |
| $D_3$ | **Social** | 1.139 | 1.049 | 2.188 | 0.090 |
| $C_5$ | Education level | 2.917 | 1.877 | 4.794 | 1.039 |
| $C_6$ | Standard of living | 3.020 | 3.453 | 6.473 | −0.433 |
| $C_7$ | Social security | 2.116 | 2.069 | 4.185 | 0.047 |
| $D_4$ | **Physical environment** | 0.993 | 1.136 | 2.130 | −0.143 |
| $C_8$ | Quality of infrastructure | 2.616 | 2.777 | 5.393 | −0.161 |
| $C_9$ | Public facilities | 2.570 | 2.893 | 5.463 | −0.323 |
| $C_{10}$ | Medical resources | 1.592 | 2.069 | 3.661 | −0.477 |
| $C_{11}$ | Public transport | 2.548 | 2.876 | 5.424 | −0.328 |
| $D_5$ | **Natural environment** | 0.832 | 0.927 | 1.759 | −0.095 |
| $C_{12}$ | Pollution index | 2.023 | 2.247 | 4.271 | −0.224 |
| $D_6$ | **Culture and creativity** | 0.965 | 1.227 | 2.191 | −0.262 |
| $C_{13}$ | Cultural and creative industries | 2.159 | 2.377 | 4.536 | −0.219 |
| $C_{14}$ | Tourism attractiveness | 2.400 | 3.430 | 5.830 | −1.031 |

The experts' opinions diverged from the centrality results in that the economics dimension was the third most influential rather than most influential dimension. In general, improvements in the economics dimension are primarily driven by improvements in the governance and society dimensions, which, in turn, influence the remaining dimensions.

The dimensions of the physical environment and culture and creativity received rather than exerted influence; however, this result does not entail that they are unimportant. Governments should be creative when formulating policies targeting these dimensions. Specifically, governments should consider how these dimensions synergize with the other dimensions in the system to formulate more holistic and thus more effective policies.

The present study used the DANP to obtain the weights of the criteria and dimensions. First, the weights of the influence matrices and supermatrices of the dimensions were integrated. Subsequently, the total influence matrix was normalized to create an unweighted supermatrix (i.e., W), which was weighted to obtain the weighted supermatrix (W*). The process of acquiring W* was iterated until the matrix converged to form a limit supermatrix.

The limit supermatrix comprised values generated after the weighted matrix converged; that is, it comprised the DANP weight values. The weights obtained here were the global weights of the criteria. By multiplying the global weights of the criteria with the weights of the respective dimensions, the local weights of the criteria were obtained.

By performing DANP matrix calculations, the weight values of all the dimensions were obtained. To determine how satisfied the participants were with Keelung's current urban competitiveness, this study distributed questionnaires on such satisfaction to five experts, two Keelung government officials, and six Keelung residents. The average satisfaction was multiplied by the weight of the dimensions to obtain the corresponding performance values. Therefore, by analyzing the questionnaire responses, we could determine the satisfaction of the experts, Keelung's public sector officials, and Keelung's residents with respect to all the dimensions and criteria. The relevant results, which are presented in Table 4, can be used to as a policy reference for Keelung.

**Table 4.** The performance of dimensions and criteria.

| Dimensions and Criteria | Local Weight (Base on DANP) | Global Weight (Base on DANP) | Performance |
|---|---|---|---|
| **Economics** | 0.173(3) | | 0.339(3) |
| Total output | 0.542(1) | 0.094 | 0.426(1) |
| Per capita income | 0.458(2) | 0.079 | 0.253(2) |
| **Governance** | 0.138(6) | | 0.274(5) |
| Government efficiency | 0.462(2) | 0.064 | 0.372(1) |
| Cross-border cooperation | 0.538(1) | 0.074 | 0.177(2) |
| **Social** | 0.167(4) | | 0.287(4) |
| Education level | 0.255(3) | 0.043 | 0.527(1) |
| Standard of living | 0.468(1) | 0.078 | 0.126(3) |
| Social security | 0.277(2) | 0.046 | 0.210(2) |
| **Physical environment** | 0.180(2) | | 0.218(6) |
| Quality of infrastructure | 0.262(3) | 0.047 | 0.471(1) |
| Public facilities | 0.272(2) | 0.049 | 0.127(3) |
| Medical resources | 0.193(4) | 0.035 | 0.173(2) |
| Public transport | 0.273(1) | 0.049 | 0.100(4) |
| **Natural environment** | 0.147(5) | | 0.407(2) |
| Pollution index | 1.000 | 0.147 | 0.407 |
| **Culture and creativity** | 0.195(1) | | 0.429(1) |
| Cultural and creative industries | 0.412(2) | 0.080 | 0.600(1) |
| Tourism attractiveness | 0.588(1) | 0.114 | 0.258(2) |
| **Total Performance** | | | 4.226 |

Note: 1. The numbers in the ( ) in Local Weight denote the ranks of local weights in dimensions and criteria. 2. The numbers in the ( ) in Performance denote the ranks of Performance.

## 5. Discussion

Most relevant studies have focused on the development and experience of core cities within examined regions, largely ignoring the complexities and heterogeneity among the other cities in the region. In the process by which metropolitan areas are formed, the power relations within the region—competition, cooperation, complementation, and division of functions—constitute essential elements that affect the development of individual cities. For Keelung, a marginalized city in northeastern Taiwan that is part of the Greater Taipei Area, rather than following existing mainstream neoliberalist practices, its urban development strategy must focus on clarifying the causes and effects of its shrinking role and functions in the contexts of globalization and regionalization. This approach will enable the city to determine its own position and development path. According to Storper and Scott (1995) [44], cities' development strategies must adhere to the following premise: cities exist in a globalized setting, and individual cities' development paths are inevitably affected by the network of cities of which they are a part.

Regarding relevant studies on Keelung, Chiu, Liu, and Hung (2020) [45] applied the boundary governance perspective in a social network analysis of the relationship and power structure between actors to identify problems in the decision-making process. Chang and Lin (2017) [46] applied the system dynamics approach in development strategies for Keelung tailored to its identity as a port city. The researchers used three situation simulations for Keelung: (1) maintenance of the current situation, (2) transition into a cruise port for tourism development, and (3) promotion of cruise travel and tourism. Yeh and Chang (2019) [47] argued that the transformation of Keelung Port's business models must be considered to overcome the challenges arising from the reductions in revenue and cargo throughput. Shiau and Chuang (2019) [48] conducted a case study of Keelung Port in examining the social construction of port sustainability indicators. These studies focused mainly on increasing Keelung Port's cargo throughput and transforming

its port businesses, as well as promoting port tourism and city network governance. A comprehensive discussion of the development of Keelung City remains lacking. As such, an urban development metric for secondary cities was developed in this study.

By using the FDM and DANP, we examined a comprehensive set of dimensions that influenced urban development and competitiveness, the effects of the dimensions on one another, and the weights of the dimensions.

The dimensions of governance (D1), society (D2), and economics (D3) were influential on the other dimensions, and the experts considered the economics and social dimensions to be relatively important dimensions. However, these two dimensions were also influenced by the governance dimension. Therefore, the governance dimension is of utmost policy importance because it implicates how well the economy and society are doing, which is the reason why the experts assessed the governance dimension to be the most important dimension for the study region's overall performance. The second most influential dimension was the social dimension, and its spillover effects on other dimensions were similar to the aforementioned ones of the governance dimension. The economics dimension was third most influential dimension and exerted and received the most influence overall.

According to the present results, the governance dimension (D2) had the strongest influence on other dimensions in the system. This is consistent with the findings of Chiu et al. (2020) [45], who reported that current governance models, which remain within a bureaucratic system network, may shift into political power models, in which decision outcomes are influenced by personal positions and power in the face of conflict. To improve existing cross-regional governance models, the authority, roles, and tasks of each department in the network must be clarified. Furthermore, old bureaucratic constraints should be abandoned. Aside from the participants of conventional decision-making networks, various actors should be involved, according to a wider range of topics, to strengthen partnerships with the private sector or third-party sectors. Through robust decentralized models and enhanced public engagement, future governance models may become effective and flexible, guiding rather than manipulating. This can drive national development. As for the economics dimension (D1) and the social dimension (D3), the findings are consistent with those of Linneman and Rybczynski (1999) [49], Hsu (2016) [50], Chang and Lin (2017) [46], and Chang, Chang, Lin, and Lin (2010) [51]: the economic activity and social vitality of a city are mutually influencing and reinforcing with regard to improving urban functions, promoting economic prosperity, and increasing residents' employment rate and sense of local identity. These factors may halt the decline of Keelung City and enable a developmental and economic rebound.

The physical environment (D4), natural environment (D5), and culture and creativity (D6) dimensions were most influenced by other dimensions. Among the aforementioned three dimensions, D6 and D5 received the most and least influence, respectively. Thus, when seeking to improve a city with respect to the aforementioned dimensions, policymakers must also consider the influence of other dimensions.

The influence throughout the network was visualized by plotting the degree of influence on the x-axis against centrality on the y-axis (Figure 3), where the middle to top and bottom parts of the plot represent the dimensions and criteria, respectively. The directions of the arrows in Figure 3 indicate the net flows of influence. For example, in the aforementioned plot, an arrow points from the governance dimension to the economics dimension because the governance dimension was more influential on the economics dimension than vice versa.

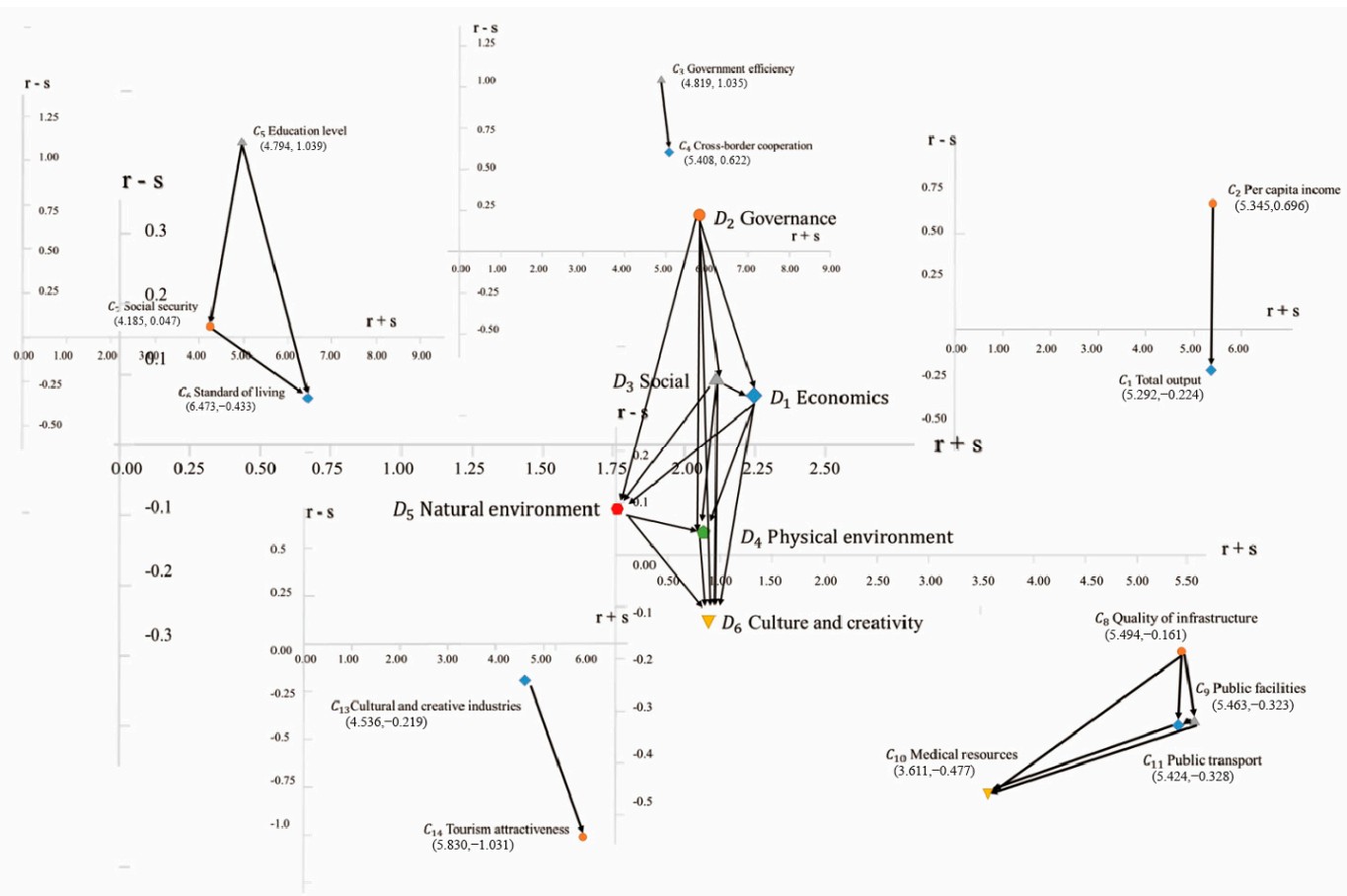

**Figure 3.** The influential network relations map of 14 criteria within 6 dimensions.

Furthermore, the participants reported the greatest to least satisfaction about the culture and creativity (0.429), natural environment (0.407), economics (0.339), society (0.287), governance (0.274), and physical environment (in particular, public transport; 0.218) dimensions, respectively. Although this result indicates that the physical environment should be a policy priority, the physical environment dimension was highly influenced by the other dimensions. Thus, Keelung's relatively poor physical environment is a symptom of a systemic problem. Therefore, policymakers should approach problems with the physical environment in a holistic, systemic manner to ensure that the root causes, rather than only the symptoms, are treated.

This study's assessment indicator framework was constructed on the basis of expert opinions. However, Cascetta et al. (2015) [52] advocated for including various groups (expert or nonexpert) to enhance decision-making. However, we believe that in addition to the framework construction stage, public participation can be increased by assessing subsequent empirical objects in future studies.

## 6. Conclusions

The present study investigated urban development and competitiveness from the perspective of a secondary region city, namely Keelung. This study conducted a literature review of various assessment methods and indices of urban competitiveness used in Taiwan and abroad. Subsequently, a system for assessing urban competitiveness was constructed to identify indices that pertain to urban development. The assessment indices were constructed using the FDM and DANP, where the effects of different dimensions on each other were determined on the basis of expert opinions. The results of this study aid policy-related decision-making in Keelung. In addition, we determined the satisfaction

of the research participants, who comprised experts and residents of Keelung, with the current situation in Keelung regarding the dimensions in the assessment system.

### 6.1. Influence of the Dimensions in the Assessment System

Indices of urban development and competitiveness were also identified using a fuzzy Delphi expert questionnaire. By using the DANP, we found that the experts considered the governance, economics, and society dimensions to be relatively important and influential on the other dimensions. By contrast, the natural environment, physical environment, and culture and creativity dimensions were considered to have relatively low importance and relatively low influence on the other dimensions. The DANP was also used to determine the weight values and the effects of the dimensions on each other. We found that governance was the most influential dimension. Thus, policymakers should focus on this dimension to formulate holistic policies even when solving problems in other dimensions.

### 6.2. Status of and Policy Prescriptions for Keelung

The participants (experts and residents) were most to least satisfied with the culture and creativity, natural environment, economics, social, governance, and physical environment dimensions, respectively, in Keelung. Within the best-performing dimension (i.e., the culture and creativity dimension), the criterion of cultural and creative industries had the highest satisfaction score and the criterion of tourist attractiveness had the lowest satisfaction score. The experts also perceived tourist attractiveness to be highly influential on city competitiveness, as reflected in its highest weight of 0.588. Policymakers can evaluate the effects of tourism by visitors from within and outside the region, enhance the value that attractions provide to tourists, and integrate tourist resources around Keelung. In addition, within the worst-performing dimension (i.e., the physical environment dimension), the public transport, public facilities, and medical resources criteria had the lowest satisfaction scores.

However, enhancing the connection of the utilities surrounding the city should also be taken into consideration under the limitation of land. Our method can uncover the influence that dimensions exert on each other and thus can help policymakers implement holistic improvements rather than piecemeal ones.

### 6.3. Assessment Framework of City Competitiveness from the Perspective of Regional Resource Integration

Keelung has declined in importance over time relative to its counterparts in the Taipei metropolitan area. Thus, we adopted a regional perspective to construct a comprehensive evaluation system. This method can be used to enhance both the internal and external conditions of the city's competitiveness.

The framework used in this study was designed according to the current status in Keelung and thus can be revised in future research. However, our results can only indicate the general direction of urban development; we did not specifically evaluate the influences of policy inputs and outputs. Future research can address this limitation.

**Author Contributions:** The individual contribution and responsibilities of the authors were as follows: Y.-H.C.: Research idea and method design, grant holder of research financing, article writing, related data collection and analysis, supervision of the research direction. Y.-Y.L.: Research data collection, literature review and analysis, article writing and formatting. All authors have read and agreed to the published version of the manuscript.

**Funding:** This research was funded by the Ministry of Science and Technology, Taiwan: MOST-105-2420-H-845-001.

**Acknowledgments:** This study is part of the "Development of the Planning Method and Governance Mechanism for Cross-boundary Integration of Cities Based on Global Competitiveness." project (Project No.: MOST-105-2420-H-845-001) administered by the Ministry of Science and Technology,

Executive Yuan. The researchers of this study would like to thank the Ministry of Science and Technology, Executive Yuan, for funding this study.

**Conflicts of Interest:** The authors declare no conflict of interest.

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
