# Peer review of "The Elaborated Assessment Framework of City Competitiveness from the Perspective of Regional Resource Integration"

_sustainability, doi:10.3390/su13115932_

Round 1
Reviewer 1 Report
This paper provides a framework for assessing urban development from the perspective of regional resource integration, and then applies this framework to a secondary city Keelung as a case study. The case study is presented in detail and clarity. The paper is also well organized. Overall, as a case study, it is a piece of good work.
However, I have some concerns about the theoretical contributions of this work. Why cannot the existing assessment framework apply to secondary cities? What are the unique aspects of those cities that warrant a new framework and method? I feel that these are critical questions to answer before acceptance.
Also, the paper needs to do a much better job in articulating the significance of the findings (especially against the existing literature). In other words, a careful and detailed comparison between the findings of this paper and those of the existing literature is required. As is, there is very limited discussion in this regard, but it is necessary to demonstrate the originality of the findings.
In addition, there are some sporadic grammar errors and typos throughout the paper. For instance, in the bottom of page 1, "The goal was to explore and analyses city positioning and function-related theories in current context ...": here 'analyses' should be changed to 'analyse'.
Author Response
Reviewer 1
This paper provides a framework for assessing urban development from the perspective of regional resource integration, and then applies this framework to a secondary city Keelung as a case study. The case study is presented in detail and clarity. The paper is also well organized. Overall, as a case study, it is a piece of good work.
Q1:However, I have some concerns about the theoretical contributions of this work. Why cannot the existing assessment framework apply to secondary cities? What are the unique aspects of those cities that warrant a new framework and method? I feel that these are critical questions to answer before acceptance.
A1:
- The levels of urban development and governance used to be measured city by city; however, with the rise of globalization, overall development is now considered from regional perspectives. Typically, when studying regional development, regions are evaluated on the basis of major cities within the region, and the roles of the remaining cities are not addressed. Most relevant studies have focused on the development and experience of core cities within examined regions, largely ignoring the complexities and heterogeneity among the other cities in the region. In the process by which metropolitan areas are formed, the power relations within the region—competition, cooperation, complementation, and division of functions—constitute essential elements that affect the development of individual cities.
- Because of the unique economic, geographic, and functional conditions of secondary cities, they display characteristics that are distinct from those of major cities, making them irreplaceable in a region. By promoting secondary cities’ development, the overall competitiveness of a region can be enhanced. However, although secondary cities provide vital resources for maintaining core cities’ daily operations and play critical roles in regional networks, secondary cities often struggle with insufficient resources and underdeveloped infrastructure, as well as investment attraction, leading to poor competitiveness and dependency on core cities. This dependency increases the likelihood of secondary cities’ marginalization in regional integrations. Therefore, when discussing city region development, in addition to examining core cities and their development and performance, the roles played by secondary cities in regional and national development should also be considered, thereby facilitating the understanding of the overall balance and complexity of the region.
- Because of these factors, current benchmarks of urban development or competitiveness are based on the evaluation of core cities. The various dimensions of urban development benchmarks may not be comprehensive, may have different weights, and may not take into account the characteristics of specific cities. For instance, agricultural cities and commercial cities require different types of development and face distinct challenges; thus, appropriate benchmark dimensions should be applied accordingly. If cities at every level were evaluated using the same benchmarks, their bases of comparison would be different; therefore, development benchmarks should be formulated to match the characteristics of cities on a given level. The urban development metrics proposed in this study target secondary cities.
Q2:Also, the paper needs to do a much better job in articulating the significance of the findings (especially against the existing literature). In other words, a careful and detailed comparison between the findings of this paper and those of the existing literature is required. As is, there is very limited discussion in this regard, but it is necessary to demonstrate the originality of the findings.
A2:
- Most relevant studies have focused on the development and experience of core cities within examined regions, largely ignoring the complexities and heterogeneity among the other cities in the region. In the process by which metropolitan areas are formed, the power relations within the region—competition, cooperation, complementation, and division of functions—constitute essential elements that affect the development of individual cities. For Keelung, a marginalized city in northeastern Taiwan that is part of Greater Taipei Area, rather than following existing mainstream neoliberalist practices, its urban development strategy must focus on clarifying the causes and effects of its shrinking role and functions in the contexts of globalization and regionalization. This approach will enable the city to determine its own position and development path. According to Storper and Scott (1995), cities’ development strategies must adhere to the following premise: cities exist in a globalized setting, and individual cities’ development paths are inevitably affected by the network of cities of which they are a part.
- Regarding relevant studies on Keelung, Chiu, Liu, and Hung (2020) applied the boundary governance perspective in a social network analysis of the relationship and power structure between actors to identify problems in the decision-making process. Chang and Lin (2017) applied the system dynamics approach in development strategies for Keelung tailored to its identity as a port city. The researchers used three situation simulations for Keelung: (1) maintenance of the current situation, (2) transition into a cruise port for tourism development, and (3) promotion of cruise travel and tourism. Yeh and Chang (2019) argued that the transformation of Keelung Port’s business models must be considered to overcome the challenges arising from the reductions in revenue and cargo throughput. Shiau and Chuang (2019) conducted a case study of Keelung Port in examining the social construction of port sustainability indicators. These studies focused mainly on increasing Keelung Port’s cargo throughput and transforming its port businesses, as well as promoting port tourism and city network governance. A comprehensive discussion of the development of Keelung City remains lacking. As such, an urban development metric for secondary cities was developed in this study.
- According to the present results, the governance dimension (D2) had the strongest influence on other dimensions in the system. This is consistent with the findings of Chiu, Liu, and Hung (2020), who reported that current governance models, which remain within a bureaucratic system network, may shift into political power models, in which decision outcomes are influenced by personal positions and power in the face of conflict. To improve existing cross-regional governance models, the authority, roles, and tasks of each department in the network must be clarified. Furthermore, old bureaucratic constraints should be Aside from the participants of conventional decision-making networks, various actors should be involved, according to a wider range of topics, to strengthen partnerships with the private sector or third-party sectors. Through robust decentralized models and enhanced public engagement, future governance models may become effective and flexible, guiding rather than manipulating. This can drive national development. As for the economics dimension (D1) and the social dimension (D3), the findings are consistent with those of Rybczynski and Linneman (1999), Hsu (2016), Chang and Lin (2017), and Chang, Chang, Lin, and Lin (2010): the economic activity and social vitality of a city are mutually influencing and reinforcing with regard to improving urban functions, promoting economic prosperity, and increasing residents’ employment rate and sense of local identity. These factors may halt the decline of Keelung City and enable a developmental and economic rebound.
- These points have been added to Section 5: Discussion.
Q3:In addition, there are some sporadic grammar errors and typos throughout the paper. For instance, in the bottom of page 1, "The goal was to explore and analyses city positioning and function-related theories in current context ...": here 'analyses' should be changed to 'analyse'.
A3: Corrected. See page 2. The goal was to explore and analyze theories of city positioning and function in the context of Keelung, which is characterized by increasingly tight regional cooperation.

Reviewer 2 Report
The paper is very interesting, since it offers a different view on how to address competitiveness analysis for a secondary city. The structure is clear and the content is also clearly explained.
Just three or four points should be revised by authors. I have included those points in the attached file.
Especially, the abstract should be revised, since it does not do justice to the interesting content that the reader will find in the paper. The discussion sections should include more "discussion", then comparisons and references to other previous works on the matter should be included.

Author Response
Reviewer 2
The paper is very interesting, since it offers a different view on how to address competitiveness analysis for a secondary city. The structure is clear and the content is also clearly explained.
Just three or four points should be revised by authors. I have included those points in the attached file.
Especially, the abstract should be revised, since it does not do justice to the interesting content that the reader will find in the paper. The discussion sections should include more "discussion", then comparisons and references to other previous works on the matter should be included.
Q1:The abstract should contain explicit reference to the study case, Keelung, as well as the main results obtained.
A1: Corrected. See page 1.
Studies on regional development have generally focused on major cities, to the neglect of minor ones. In this research, a secondary city in Taiwan, namely Keelung, was selected as a case study for urban development assessment from the perspective of regional resource integration. This study combined the decision-making trial and evaluation laboratory (DEMATEL) method and analytic network process (ANP) to determine how dimensions influenced each other in Keelung and what their weights were. We used six dimensions that comprised 14 criteria. The adopted dimensions were economics, governance, society, physical environment, natural environment, and culture and creativity. On the basis of the DEMATEL-based ANP, experts considered the governance, economics, and society dimensions to be highly important and influential on the other dimensions. In elucidating the relationships between dimensions, our method allows policymakers to formulate holistic solutions rather than piecemeal ones. The satisfaction of experts and residents (who have expertise and considerable living experience in Keelung, respectively) with the current situation in Keelung regarding the dimensions and criteria was also determined.
Q2:Please, provide long term/concept, since it is the first appearance of this abbreviation in the text.
A2: Corrected. See 3. Methods. This study used multiple criteria decision-making, the fuzzy Delphi method (FDM), and the decision-making trial and evaluation laboratory (DEMATEL)-based analytic network process (ANP) to explore the urban development strategy of a secondary city.
Q3:Please, provide some reference that arghe for these advantages.
A3: Corrected. See 3.1. The Fuzzy Delphi Method.
The FDM evolved from combining the conventional Delphi method and fuzzy set theory. The Delphi method is tedious and time-consuming to implement, and the questions in the questionnaire might be semantically confusing, thus, causing erroneous responses; as such, subsequent scholars improved the drawbacks of the conventional Delphi method. Ishikawa et al. (1993) incorporated fuzzy set theory into the Delphi method to overcome these problems and employed the concepts of cumulative frequency distribution and fuzzy integral to transform the expert opinions into fuzzy numbers, through which the FDM was developed. Jeng (2001) employed the double triangular fuzzy technique and grey zone test method to examine and integrate expert opinions and to reduce the frequency of repeated questionnaire surveys. Since these approaches are more objective and reasonable than the single triangular fuzzy technique for obtaining the geometric mean, the present study developed the relevant criteria and indices for assessing the secondary city.
Q4:The table is cut, but I guess this will be solved in the final editing process.
A4: Corrected. See Table 3. The sum of influences given and received on dimensions and criteria.
Q5:The discussion sections, in its current form, is repeating or completing aspects related to the results obtained, but it should contain a discussion based on other results from others authors and works.
A5:
- Most relevant studies have focused on the development and experience of core cities within examined regions, largely ignoring the complexities and heterogeneity among the other cities in the region. In the process by which metropolitan areas are formed, the power relations within the region—competition, cooperation, complementation, and division of functions—constitute essential elements that affect the development of individual cities. For Keelung, a marginalized city in northeastern Taiwan that is part of Greater Taipei Area, rather than following existing mainstream neoliberalist practices, its urban development strategy must focus on clarifying the causes and effects of its shrinking role and functions in the contexts of globalization and regionalization. This approach will enable the city to determine its own position and development path. According to Storper and Scott (1995), cities’ development strategies must adhere to the following premise: cities exist in a globalized setting, and individual cities’ development paths are inevitably affected by the network of cities of which they are a part.
- Regarding relevant studies on Keelung, Chiu, Liu, and Hung (2020) applied the boundary governance perspective in a social network analysis of the relationship and power structure between actors to identify problems in the decision-making process. Chang and Lin (2017) applied the system dynamics approach in development strategies for Keelung tailored to its identity as a port city. The researchers used three situation simulations for Keelung: (1) maintenance of the current situation, (2) transition into a cruise port for tourism development, and (3) promotion of cruise travel and tourism. Yeh and Chang (2019) argued that the transformation of Keelung Port’s business models must be considered to overcome the challenges arising from the reductions in revenue and cargo throughput. Shiau and Chuang (2019) conducted a case study of Keelung Port in examining the social construction of port sustainability indicators. These studies focused mainly on increasing Keelung Port’s cargo throughput and transforming its port businesses, as well as promoting port tourism and city network governance. A comprehensive discussion of the development of Keelung City remains lacking. As such, an urban development metric for secondary cities was developed in this study.
- According to the present results, the governance dimension (D2) had the strongest influence on other dimensions in the system. This is consistent with the findings of Chiu, Liu, and Hung (2020), who reported that current governance models, which remain within a bureaucratic system network, may shift into political power models, in which decision outcomes are influenced by personal positions and power in the face of conflict. To improve existing cross-regional governance models, the authority, roles, and tasks of each department in the network must be clarified. Furthermore, old bureaucratic constraints should be abandoned. Aside from the participants of conventional decision-making networks, various actors should be involved, according to a wider range of topics, to strengthen partnerships with the private sector or third-party sectors. Through robust decentralized models and enhanced public engagement, future governance models may become effective and flexible, guiding rather than manipulating. This can drive national development. As for the economics dimension (D1) and the social dimension (D3), the findings are consistent with those of Rybczynski and Linneman (1999), Hsu (2016), Chang and Lin (2017), and Chang, Chang, Lin, and Lin (2010): the economic activity and social vitality of a city are mutually influencing and reinforcing with regard to improving urban functions, promoting economic prosperity, and increasing residents’ employment rate and sense of local identity. These factors may halt the decline of Keelung City and enable a developmental and economic rebound.
- These points have been added to Section 5: Discussion.

Reviewer 3 Report
This is an interesting paper which presents a methodology which can be widely replicated, and therefore it is worthy of publication. However, the text needs to be completely revised by a native English speaker, because in its present form it is very hard to read, as there are numerous grammatical errors.
Author Response
已更正。

Reviewer 4 Report
- In my opinion, the old publication template was used
- Please also check the methods of quoting in the test - the first citation is in position 20
Author Response
Corrected.

Round 2
Reviewer 3 Report
The English language has been revised and the paper is consequently much improved.